# Molecular Changes Associated with Suicide

**DOI:** 10.3390/ijms242316726

**Published:** 2023-11-24

**Authors:** Daniela Navarro, Marta Marín-Mayor, Ani Gasparyan, María Salud García-Gutiérrez, Gabriel Rubio, Jorge Manzanares

**Affiliations:** 1Instituto de Neurociencias, Universidad Miguel Hernández-CSIC, Avda de Ramón y Cajal s/n, San Juan de Alicante, 03550 Alicante, Spain; dnavarro@umh.es (D.N.); agasparyan@umh.es (A.G.); maria.ggutierrez@umh.es (M.S.G.-G.); 2Redes de Investigación Cooperativa Orientada a Resultados en Salud (RICORS), Red de Investigación en Atención Primaria de Adicciones (RIAPAd), Instituto de Salud Carlos III, MICINN and FEDER, 28029 Madrid, Spain; grubiovalladolid@gmail.com; 3Instituto de Investigación Sanitaria y Biomédica de Alicante (ISABIAL), 03010 Alicante, Spain; 4Instituto de Investigación i+12, Hospital Universitario 12 de Octubre, 28041 Madrid, Spain; marta.marin@salud.madrid.org; 5Department of Psychiatry, Complutense University of Madrid, 28040 Madrid, Spain

**Keywords:** suicide, molecular alteration, risk factor, gene, epigenetic change

## Abstract

Suicide is a serious global public health problem, with a worrying recent increase in suicide rates in both adolescent and adult populations. However, it is essential to recognize that suicide is preventable. A myriad of factors contributes to an individual’s vulnerability to suicide. These factors include various potential causes, from psychiatric disorders to genetic and epigenetic alterations. These changes can induce dysfunctions in crucial systems such as the serotonergic, cannabinoid, and hypothalamic–pituitary–adrenal axes. In addition, early life experiences of abuse can profoundly impact an individual’s ability to cope with stress, ultimately leading to changes in the inflammatory system, which is a significant risk factor for suicidal behavior. Thus, it is clear that suicidal behavior may result from a confluence of multiple factors. This review examines the primary risk factors associated with suicidal behavior, including psychiatric disorders, early life adversities, and epigenetic modifications. Our goal is to elucidate the molecular changes at the genetic, epigenetic, and molecular levels in the brains of individuals who have taken their own lives and in the plasma and peripheral mononuclear cells of suicide attempters and how these changes may serve as predisposing factors for suicidal tendencies.

## 1. Introduction

Suicide is the cause of more than 700,000 deaths each year. Shockingly, in 2019, suicide was the fourth leading cause of death among people aged 15 to 29, according to the World Health Organization (WHO) [1]. Suicide is a severe global problem that is preventable. The need for a comprehensive, multisectoral suicide prevention strategy is paramount. However, it is crucial to recognize that many factors influence an individual’s risk of suicide. 

The link between suicide and mental disorders is well established. Psychiatric disorders such as major depression and schizophrenia are among the most critical risk factors for suicide. In addition, early experiences of sexual and psychological abuse are strongly related to suicidal behavior [2]. Both chronic and acute stressful situations can increase suicidal behavior [2]. In addition, many suicides occur impulsively and unexpectedly in times of crisis due to a lack of ability to cope with stressful situations. Suicide victims show several alterations at both the central and peripheral levels. For example, the prefrontal cortex (PFC) of suicide victims shows an increase in the inflammatory interleukins IL-1, IL-6, and tumor necrosis factor alpha (TNF-alpha) and a decrease in brain-derived neurotrophic factor (BDNF) [3]. The abnormal functioning of the cortisol-stress axis and the abnormal interactions of the inflammatory system are significant risk factors for stress-related suicidal behavior [4,5]. The potential markers of suicidal behavior include those related to changes in the inflammatory pathways associated with hypothalamic—pituitary—adrenal (HPA) axis hyperactivity. Several molecular changes in cells associated with neuroinflammation and the immune system have been identified in the brains of suicide victims, including astroglia and microglia [6].

Suicidal behavior is not solely attributed to stressful situations or psychiatric disorders that cause molecular changes; genetic factors also play a role. Seven genes have been identified that are differentially expressed in suicide but not mood disorders [7]. Familial genetic inheritance may be a cause of suicidal behavior. It has been reported that monozygotic twins present a higher concordance for suicidal behavior than dizygotic twins [8,9,10]. In addition, alterations in genes encoding proteins involved in serotonergic neurotransmission have been associated with suicidal behavior [10]. Furthermore, epigenetic changes such as DNA methylation were detected in suicide cases compared to non-psychiatric controls in both the PFC and cerebellum [11]. 

Therefore, suicidal behavior may be multifactorial, involving a complex interplay of genetic, epigenetic, psychiatric, psychological, and environmental factors [12]. These changes may disrupt stress control, neuroplasticity, and neurotransmission systems [2,3]. However, much remains to be understood about the biological factors underlying the pathophysiology of suicide. 

In this review, we have considered the molecular changes in suicide attempters and completers. While we examine the risk factors associated with suicidal behavior in psychiatric disorders and early life adversity, our primary goal is to highlight the most well-studied factors that may predispose individuals to suicide. In addition, we aim to elucidate the molecular changes at the genetic, epigenetic, and system levels observed in the brains of suicide completers, in the plasma and peripheral blood cells of the suicide attempters and how these changes may be a risk factor for suicidal behavior.

## 2. Mental Disorders as Risk Factors for Suicide

Suicidality can be conceptualized as a continuum from thoughts to actions, with intensity from mild to severe, which includes, in ascendant order, suicidal ideation, suicide planning, suicide attempts, and suicide deaths [13]. It is well established that, compared to individuals without a psychiatric disorder, those with severe mental illness are more likely to commit suicide. This section will review different mental disorders as risk factors for suicide.

### 2.1. Bipolar Disorder

Multiple psychiatric disorders have been associated with an increased risk of suicidal behaviors. However, bipolar disorder (BD) is the one that confers the highest risk of suicide. Both suicide attempts and completed suicides are significantly more prevalent in patients with BD than in the general population. The risk for suicide among BD patients is about 60 times greater than in the general population. Patients with BD are also the most likely to engage in repeated suicide attempts among all psychiatric disorders. Approximately 25–50% of patients with BD will commit at least one suicide attempt during their lifetime, and 10–20% will complete suicide. It is essential to highlight that suicide attempts in BD patients tend to be associated with greater lethality, as confirmed by the ratio of suicide attempts to completed suicides, which is much lower in BD patients (3:1) compared to the general population (35:1). Therefore, patients with BD die by suicide 20 to 30 times more frequently than their peers in the general population [14,15,16,17,18,19,20,21,22,23,24]. 

Several sociodemographic, clinical, genetic-neurobiological, and treatment risk factors have been related to suicidality in BD patients. Regarding sociodemographic factors, most studies point out that women with BD are significantly more likely than men with BD to attempt suicide. On the contrary, men with BD are more likely to attempt suicide by violent methods and have significantly higher absolute rates of suicide deaths [16,20,23,24]. Considering age, BD patients who attempt suicide are significantly younger than non-attempters, but older age is associated with a higher suicide lethality [16,17,20]. Finally, other sociodemographic factors proposed for the association of BD and suicidality include divorced or single marital status, inadequate social support, low socioeconomic status, a lower level of education, stressful life events, and a higher mean altitude of residence [16,17,21,22,23,25].

An earlier onset of the disease is associated with an increased risk of suicide attempts in BD patients [14,16,20,21,22,23,26]. Moreover, BD patients who committed more than one suicide attempt showed an even significantly lower age of onset of the disease [26]. Different studies have also reported that BD individuals who attempt suicide have a significantly longer duration of the disorder, independent of gender. Furthermore, a longer duration of the disorder has been positively correlated with a higher lethality of suicide attempts [15]. Rapid cycling (having at least four affective episodes within one year) has also been strongly associated with an increased lifetime risk of suicidal behavior [23,24]. Other clinical variables associated with an increased risk of suicidality in BD patients are a more significant number and severity of affective episodes, especially depressive episodes and symptoms, having treatment-resistant depression, hopelessness, psychotic features, atypical depressive features, agitated depression, and dysthymic or cyclothymic temperaments [15,17,20,22,26]. 

Comorbidity with other psychiatric disorders and substance use disorders (SUD) increases the likelihood of suicidal behavior in BD individuals. It has also been reported that there are consistent and positive associations between comorbid alcohol and substance use in patients with BD and suicide attempts, regardless of age and gender. Finally, suicide attempters exhibit cognitive difficulties and impaired decision-making, leading to using suicidal behaviors and/or alcohol and drugs as a maladaptive way of managing painful feelings [15,21,23,27]. Other personality traits related to suicidality in BD patients include impulsivity, aggressiveness, difficulties in emotion regulation, a low tolerance to stress, extreme attributional style, lower social skills, and self-directedness [15,28]. 

Finally, treatment risk factors should also be considered. It has been suggested that antidepressants may increase the risk of suicidality in BD patients, primarily when they associate dysphoric agitation, anger, restlessness, irritability, or insomnia, or, during the mood episode, mixed features are predominant, dysphoric mania or agitated depression. This antidepressant-increased suicide risk has been observed for younger patients, whereas, in older patients, antidepressants could show a protective effect against suicide [20]. On the contrary, lithium treatment has been proposed to show a protective effect over suicidality. Several studies have found a reduced suicide risk in BD patients with lithium [20]. 

The increased likelihood of suicide among BD patients is related to specific genetic and neurobiological correlates. Suicide attempts among individuals with BD are associated with periventricular white matter and deep white matter hyperintensities [15]; lower grey matter volumes in regions including the ventrolateral prefrontal cortex and the dorsolateral prefrontal cortex (DLPFCx) [29], the right orbitofrontal cortex, the right hippocampus (HIPP) and bilateral cerebellum [15,30]; lower left orbitofrontal cortical thickness and white matter fractional anisotropy [15,25]; and a lower volume in the left ventrolateral prefrontal cortex, right DLPFCx (involved in executive function, particularly voluntary emotional regulation) and bilateral HIPP [31]. In addition, a longitudinal study that included patients with mood disorders (major depressive disorder [MDD] and BD) found lower grey matter volume in prefrontal regions at baseline, including the ventral prefrontal cortex, among youth who went on to make suicide attempts within three years of follow-up [31]. 

Other biological factors associated with an increased risk for suicidality in BD patients include altered functioning of the HPA-axis with significantly higher levels of corticotropin-releasing hormone (CRH) and bedtime salivary cortisol [15,17], and higher serum testosterone concentrations, lower serum cholesterol, triglyceride, and n-6 arachidonic acid [15]. In a large-scale biomarker discovery study, 16 candidate proteins were identified as being associated with prospective suicide attempts, with three having a robust association: uteroglobin (SCGB1A1), annexin A10 (ANXA10) and centrin 2 (CETN2). These proteins respond to hormonal and steroid activity, immune system processes, cellular growth, and metabolism [32] (Figure 1).

Genetic variations play an essential role in the association between BD and suicidality. Several gene polymorphisms, such as those associated with tryptophan hydroxylase 1 (TPH1) and tryptophan hydroxylase 2 (TPH2), were found to be related to suicide attempts of high lethality and completed suicides, respectively [17]. In addition, serotonin polymorphisms and a low expression of the serotonin receptor 2A (5-HT2A) RNA in the prefrontal cortex have been strongly associated with suicidal behavior in BD patients [15,17]. It has also been suggested that the expression of BDNF is significantly reduced in individuals who commit suicide independently of the psychiatric disorder [17]. Genome-wide association studies (GWAS) on suicide attempts in BD patients have found a trend of association with rs2767403, an intron variant of the Aminopeptidase O (AOPEP) gene, which encodes a metalloprotease linked to the renin-angiotensin system, and strong support for the chromosome 4 locus in LOCI105374524, a non-coding RNA of the glucosylceramidase beta 3 (GBA3) gene [33]. NOTCH and gamma-aminobutyric (GABA)-ergic signaling were also associated with specific related phenotypes [34]. Finally, polygenic scores for externalizing behaviors, smoking, suicide attempts, and MDD have been associated with suicide attempts and ideation in BD patients [35]. 

### 2.2. Major Depressive Disorders

Major depressive disorders (MDD) are also strongly associated with suicidality. Mortality from suicide in individuals with MDD is more than 20 times higher compared with the general population. A meta-analysis of observational studies found a pooled lifetime prevalence of suicide attempts of 31% [95% confidence interval (CI) 27–34%], a 1-year prevalence of 8% (95% CI 3–14%) and a 1-month prevalence of 24% (95% CI 15–34%) in individuals with MDD [19]. More recently, another meta-analysis has found that in patients diagnosed with MDD, the prevalence of suicide ideation was 37.7%, whereas the prevalence of suicide plans was 15.1% [36]. 

Several risk factors have been proposed for the association of MDD and suicidality. These risk factors include male gender, depression severity, age of onset and course of depression, a family history of psychiatric disorders, previously attempted suicide, comorbid psychiatric disorders such as anxiety disorders, and abuse and violence during childhood [37,38,39,40]. In addition, specific symptoms of depression, such as hopelessness, self-blame, irritability, insomnia and bipolar features, have also been associated with an increased risk of suicidality in individuals with MDD [39,40]. 

The increased risk of suicidality in patients with MDD has been associated with inflammatory changes such as lowered BDNF or increased interleukin 1ß (IL-1ß), interleukin-6 (IL-6), interleukin-13 (IL-13), tumor necrosis factor α (TNF-α), C reactive protein (CRP) and C-C motif chemokine ligand 2 (CCL2), dysfunctions of the HPA-axis, with overproduction of CRH and elevated glucocorticoid production, and structural and functional brain changes [36,41,42] (Figure 1). Recent GWAS analyses have shown that genetic liability for MDD increases the risk for suicide attempts, as suicide-attempters with MDD, BD, and schizophrenia had higher polygenic risk scores for MDD than non-attempters [33].

### 2.3. Anxiety Disorders

Anxiety disorders (AD) have been associated with suicide behavior in the general population and individuals with other psychiatric comorbidities. Findings regarding the relationship between anxiety disorders and the risk for suicidality have been inconsistent, with some studies concluding that anxiety disorders increase the risk of suicide [43] and others failing to find this significant association [44]. Within those studies with negative results, authors have suggested that specific anxiety symptoms (e.g., hypochondriasis) could be protective against suicide if associated with fear of death or illness [44]. Anxiety sensitivity, defined as the extent to which an individual fears anxiety-related sensations because of their misinterpretation, has also been found to be associated with suicidal ideation and global suicide risk [45]. 

Several modulators and mediators of the association between anxiety disorders and suicide behavior have been proposed. Some studies conclude that women are at a higher risk of suicide ideation and attempts, while others consider the opposite, although there is a certain consensus about the fact that men are more likely to commit suicide than women [46,47,48]. Furthermore, factors related per se to the disorder, such as the severity of symptoms, age of onset, mean length of the disease, and history of lifetime suicide attempts, are correlated with suicidal behavior [47,49,50].

Post-traumatic stress disorder (PTSD) is more strongly associated with suicide attempts [51], although other anxiety diagnoses have been proposed to predict suicide. In this way, panic disorder, social anxiety disorder, and generalized anxiety disorders have been proposed as significant risk factors for suicide. Finally, several reviews and meta-analyses have confirmed that, contrary to the traditional thought that obsessive compulsive disorders (OCD) are unrelated to elevated suicide risk, individuals with OCD are at an increased risk for suicide ideation and suicide attempts, so at least one patient out of ten with OCD has a lifetime suicide attempt, and nearly half of them suicidal ideation [49,52].

Although the causal role of anxiety in suicidality remains unclear, anxiety disorders and suicidality could share genetic risk and common neurobiological mechanisms (dysregulation of inflammatory responses, alterations in the HPA-axis and the CRH, structural and functional abnormalities in brain regions related to reward and affect) or mutual risk factors (childhood maltreatment has been linked to both, anxiety disorders and suicidality) [47] (Figure 1). Regarding psychological theories of suicide, anxiety is implicated in many suicide theories. It has also been proposed that avoidance strategies associated with anxiety disorders exacerbate psychological distress, which could, in turn, place patients with anxiety in a more vulnerable stage of suicidal behavior, as suicidality has been proposed to be an escape-based response from aversive self-awareness and negative emotions [43].

Genetic studies of anxiety disorders associated with suicidal behavior are scarce. However, a 22-year longitudinal study of 1255 Canadian individuals showed that the *N1-acetyltransferase* (*SAT1*) and *SAT1, spermine synthase* (*SMS*) polymorphisms are associated with anxiety and suicide attempts [53].

### 2.4. Psychotic Disorders, and Schizophrenia

Psychotic disorders are one of the psychiatric disorders more frequently associated with suicidality. A recent meta-analysis found that, among psychotic disorders, schizoaffective disorder is associated with the highest risk of suicide attempt and completed suicide, followed by schizophrenia and first psychotic episodes, with the risk of suicide-related to delusional disorders much lower [54]. Focusing on schizophrenia, the relative risk of suicide is 12 times higher in patients diagnosed with schizophrenia than those without this diagnosis. It has been estimated that 10–22% of patients diagnosed with schizophrenia complete suicide with the result of death, 20–40% of patients attempt suicide, and up to 40% of patients with a psychotic first episode experience persistent suicide ideation [55]. Several meta-analyses have been recently conducted, and they conclude that in schizophrenia, the lifetime prevalence of suicide ideation is 19.22% to 34.5%, whereas the lifetime prevalence of suicide attempts and self-harm is 14.6% to 26.8% [55,56,57,58]. 

Within the subtypes of schizophrenia, the paranoid type has been associated with a higher risk of suicidality [59]. Moreover, comorbid depression, panic attacks, alcohol and/or substance use disorders, suicide ideation and plans and a history of suicide attempts are essential contributors to suicidality. Finally, non-adherence to antipsychotic treatment also increases the risk of suicide in individuals diagnosed with schizophrenia [28,59,60,61].

As in other psychiatric disorders, it has been suggested that suicidal behavior in schizophrenia has a neurobiological basis. The hyperactivity of the HPA-axis would result in glucocorticoid neurotoxicity and subsequently in the tissue damage of different brain regions, and this could be the underlying causal mechanism of suicidality in individuals with schizophrenia. Volumetric abnormalities in the limbic system structures have been shown to discriminate between individuals with schizophrenia who commit suicide and those who do not. Individuals with schizophrenia who commit suicide show significantly lower cerebrospinal fluid concentrations of the serotonin metabolite 5-hydroxy-indol acetic acid (5-HIAA) and a blunted prolactin secretion in response to the D-fenfluramine test (Figure 1). 

Genetic variants in the serotonin receptor could play a role in the association between schizophrenia and suicide. The decreased expression levels of *2-glutamate-related genes*, *glutamate-ammonia ligase* and glial high-affinity *glutamate transporter member 3* (*SLCIA3*) have also been found in individuals with schizophrenia who commit suicide. In addition, polymorphisms in *adrenoceptor alpha 2B* (*ADRA2B*) are associated with suicidality among patients diagnosed with schizophrenia [59,61].

### 2.5. Borderline Personality Disorder

A growing body of research has shown that personality disorders, especially borderline personality disorder (BPD), are associated with a high risk of suicidal ideation, suicide attempts and completed suicide [62]. Meta-analytic results confirm a 52-fold increase in suicide rates among BPD individuals compared with the general population. At least 75% of patients with BPD attempt suicide. BPD patients have a mean of three lifetime suicide attempts, primarily by overdose. It has been reported that suicide occurs in up to 10% of BPD patients in retrospective studies and 3–6% in prospectively followed cohorts. In addition, nearly 6% of patients with BPD die due to suicide [63,64]. 

Emotional dysregulation, identity disturbances, interpersonal difficulties, and a lack of control over impulses make BPD patients more likely to turn to suicidal behaviors, so the frequency of suicide attempts is strongly associated with these traits. In addition, suicide attempts in BPD patients usually occur following stressful life events as a wish to escape [62,63,64]. 

On the other hand, BPD is associated with an increased likelihood of suicidality when it appears comorbid with other psychiatric disorders such as affective and anxiety disorders, eating disorders and SUD [62,65] (Figure 1). BPD features independently predict the risk of clinically significant suicidal ideation and the risk of suicide attempts [65]. 

Two genetic studies suggest an association between BPD and polymorphisms in the serotonergic system, such as *TPH-1* (limiting enzyme for 5-HT biosynthesis) in females and *serotonin transporter-linked promoter region* (*5-HTTLPR*) in males [66,67]. Interestingly, one found an association between the *catechol-O-methyltransferase COMT Val158 Met* single nucleotide polymorphism (SNP), the *5-HTTLPR S/L* and BPD [66].

### 2.6. Substance Use Disorders

Individuals with substance use disorders (SUD) are also at high risk for suicidality. In a meta-analysis conducted by Poorolajal et al., a significant association between SUD and suicidal ideation (OR = 2.04; 95% CI: 1.59, 2.50), suicide attempts (OR = 2.49; 95% CI: 2.00, 2.98) and suicide deaths (OR = 1.49; 95% CI: 0.97, 2.00) was reported [68]. More recently, a meta-analysis of longitudinal studies concluded that heavier alcohol use predicted an increased risk of death by suicide. Specifically, alcohol use was associated with a 94% increase in the risk of death by suicide [69]. 

SUD and suicidality influence each other in both adolescent and adult populations [70]. Several hypotheses have been developed to explain this bidirectional association: (1) the secondary psychiatric disorder hypothesis supports the idea that SUD leads to suicidality through an increase in psychological distress and impulsivity, depressive-induced symptoms, as well as a decrease in effective coping strategies and problem-solving, the quality of social relationships and/or decreased academic or working performance; (2) the secondary SUD hypothesis postulates that suicidality leads to SUD through increased coping motives for substance use; (3) the bidirectional hypothesis posits SUD and suicidality increases vulnerability each other; and (4) the common factor hypothesis points out that external factors, such as traumatic life events or impulsivity are common to both SUD and suicidality and explain their co-occurrence [70].

Some genetic studies have attempted to correlate gene alterations in patients with SUD who died by suicide. A recent study found that the *5-HTTLPR* plays a role in the association between drug use and suicidal behavior in a person with BD [71]. However, in a Slovenian population study, no association was found between polymorphism in the *5HTR* gene and transporter gene and alcohol-related suicide [72]. Patients with opioid and cocaine use disorders who attempted suicide were associated with a single nucleotide polymorphism (in the *BDNF* gene, rs7934165), suggesting a common BDNF-related pathophysiology of suicide attempts and SUD [73]. Brain gene expression patterns of suicide completers and comorbid with SUD were reported a few years ago. The authors conclude that the interaction between suicide and SUD leads to a unique expression profile. In particular, alterations in the expression of genes involved in glial differentiation and glutamatergic neurotransmission were observed in suicide with SUD [74]. 

## 3. Childhood Abuse and/or Neglect as a Risk Factor for Suicide

Childhood maltreatment is any sexual, physical, or emotional abuse and/or neglect that results in actual or potential harm that affects the child’s physical or mental health [75]. The prevalence of childhood maltreatment among the general population has been estimated to be up to 30% [75,76].

A growing body of research has found a strong association between childhood maltreatment and suicide behavior during adulthood [75,77,78] (Figure 2).

In the National Epidemiological Survey on Alcohol and Related Conditions (NESARC), childhood physical and sexual abuse was associated with suicide attempts in 2.64% of survey respondents [79]. Furthermore, this study found that childhood maltreatment was significantly associated with an increased risk of suicide at an earlier age. It has also been proposed that childhood maltreatment could additively impact suicidality [80]. In addition, in a study that examined the relationship between childhood maltreatment and prospective suicidal ideation in a cohort of youths, it was concluded that childhood maltreatment predicted suicidal ideation independently of previous suicidal ideation and depression symptoms severity [81].

Multiple meta-analyses [75,78,82,83,84,85,86,87] have studied the relationship between childhood maltreatment and the risk of suicidality in both infant adolescent and adult populations, concluding that childhood abuse and/or neglect is associated with subsequent increased risk for suicide behavior. These studies have found that children, adolescents, and adults exposed to abuse or neglect during childhood have a 2 to 4-fold increased risk for suicidality, including suicide ideation, suicide plans, and suicide attempts. Specifically, early sexual abuse has been suggested to be the type of childhood maltreatment more strongly associated with suicidal behavior, showing an additional direct effect over other forms of maltreatment [79]. Moreover, early sexually abused males would be at a higher risk for suicide attempts than females, although early sexual abuse is more prevalent in the female population [88]. 

Different mediators and moderators of the association between childhood abuse and/or neglect and suicide behavior have been analyzed. They include age [78,86], gender [83,84], the presence of a psychiatric [89] or substance use [90], comorbidity, the characteristics of the abuser and the maltreatment experience, perceived social support and specific psychopathological symptoms such as impulsivity, dissociation, personality dysfunction, emotional dysregulation, depression, uncontrollable worry, low self-esteem, a sense of hopelessness, and feelings of worthlessness and guilt [90,91,92]. 

Several biological mechanisms by which childhood maltreatment increases the risk for suicide behavior have been proposed, including inflammation and other dysfunctions of the immune system, alteration in the HPA-axis, structural and functional brain changes, and genetic and epigenetic factors [93]. 

Early-life abuse causes the persistent hyperactivity of CRF and a markedly sensitized pituitary-adrenal and autonomic response, resulting in heightened stress responses. Individuals with a history of childhood maltreatment and suicide attempters exhibit lower plasmatic cortisol levels [94,95]. 

Childhood maltreatment has been associated with structural and functional neuroimaging changes that include lower grey matter volumes, decreased thickness, and hypoactivation in the ventral, medial, and dorsal prefrontal cortex, including the orbitofrontal and anterior cingulate cortices, HIPP, insula, and striatum. An association with decreased within and between white matter structural integrity regions has been described [84,85,93,96]. These alterations can lead to cognitive impairments, the disruption of emotional regulation, and an increased risk of suicidality. 

Genetic and epigenetic mechanisms, especially related to HPA-axis genes, seem to interact with childhood abuse and/or neglect, increasing the risk of suicide behavior. Specifically, polymorphism of the following genes: *corticotropin-releasing hormone receptor 1* (*CRH1*) and *2* (*CRH2*), *FK506 binding protein 5* (*FKBP5*), and *corticotropin-release hormone binding protein* (*CRHBP*), could constitute specific mediators between childhood maltreatment and suicidality. *FKBP5* (rs3800373, rs9296158, and rs306078) interactions with childhood trauma could increase the risk of suicide attempt, whereas variations in *CRH1* (rs110402 and rs242924) would moderate the effect of children maltreatment on cortisol responses to Dexamethasone/CRH tests. Adults with a history of childhood maltreatment who carry the short-arm allele of the serotonin transporter promoter polymorphism are at a higher risk for suicidal ideation and attempts than homozygotes with two long-arm alleles. In addition, GWAS analyses have identified 14 independent loci for childhood maltreatment [93,97].

## 4. Hypothalamic-Pituitary-Adrenal Axis and Suicide

Early life stress has been associated with an aberrant HPA-axis activity in adulthood [98]. Some findings suggest that HPA-axis dysregulation could be related to endogenous forms of depression with greater suicide risk (Figure 2). However, the exact relationship between early life adversities, neuroendocrine changes later in life, and suicide remains to be determined.

Childhood trauma increases the risk of suicidal ideation and has been linked to HPA- axis dysregulation. The dexamethasone suppression test (DMS) has been used to evaluate the HPA-axis activity. Although this test presents low sensitivity and specificity, it has been suggested as a predictive marker of suicide behavior. The correlation between DMS and suicidal ideation has been analyzed in adolescent girls with a history of depression or depression plus self-injury. Interestingly, a reduction in post-DMS cortisol blood concentrations could be related to self-injury in depressed patients, suggesting a critical dysregulation of the HPA-axis negative feedback [99]. Therefore, a blunted cortisol reactivity to stress could be associated with an increased risk of suicidal behavior. 

In a study conducted with 160 participants (100 women), a correlation between suicidal attempts or ideation, cortisol levels in saliva, and childhood trauma was evaluated. The authors found that the highest levels of childhood trauma were reported in suicidal attempts in more than 78% of cases. In addition, childhood trauma was a predictor of an impaired cortisol reactivity to stress and lower cortisol concentrations were associated with higher levels of trauma in patients with a suicidal history [100]. These results were analyzed in a general population study, investigating the correlation between childhood trauma, suicidal ideation or behavior later in life and basal cortisol urine levels. A total of 1094 participants were recruited, with a history of early life difficulties (including childhood and adolescence), suicidal ideation and attempt, and 24-h urine for cortisol concentration measurements. The results indicated that early stress exposure, especially during childhood, increases the risk of suicide by around 21%. However, early adversities during childhood and adolescence are equally crucial for suicidal ideation.

Interestingly, no association was found between suicidal ideation/behavior, early stress, and urine cortisol levels [101]. These findings are not consistent with those reported in previous studies on the HPA-axis dysregulation and suicide. These discrepancies could be primarily associated with the different types of samples used to quantify cortisol concentrations (urine, blood, or saliva) or, less importantly, with the moment of sample extraction. 

An interesting aspect is that exposure to early life adversities could have a transgenerational impact affecting stress response in offspring. A sample of women with a history of major depressive disorder and childhood physical and sexual abuse has been used to analyze maternal and infant salivary cortisol levels during a laboratory stress paradigm at six months postpartum. Maternal early abuse relates to lower cortisol levels and reduced baseline cortisol in their offspring. A history of maternal abuse increases cortisol levels after laboratory stress exposure, and early abuse with comorbid PTSD in mothers induced a pronounced increase in cortisol levels in infants [102]. 

To further explore the potential relationship between the HPA-axis, suicide, and early life stress, the protein expression of CRH, FK506-binding protein (FKBP5), brain-derived neurotrophic factor (BDNF), and glucocorticoid receptors (GR) were analyzed in a study conducted using postmortem human prefrontal cortex (BA9) and anterior cingulate cortex (BA24) samples. The authors found a significant interaction between suicide and early life adversity in BA24, related to suicide behavior but not early life adversity. Therefore, this postmortem study revealed that brain alterations on targets closely related to the HPA-axis are not directly associated with suicidal behavior. Still, the BDNF protein levels could disclose a history of early life stress [103]. *CRH* gene expression levels were also analyzed in other brain regions, such as the frontopolar, dorsomedial, and ventrolateral PFC, from suicide victims (N = 12) and controls (N = 12).

Interestingly, *CRH* levels were increased in suicide victims versus controls in the frontopolar and dorsomedial PFC but not in the ventrolateral PFC. This increase was accompanied by a statistically significant reduction in *CRH1* but not the *CRH2* receptor gene expression in the frontopolar region of the PFC. [104]. The DLPFCx and anterior cingulate cortex (ACC) obtained from two different brain banks were evaluated in another study by Zhao and colleagues. The first one included samples from young depressed patients who died by suicide (MDD-S) and depressed patients who died from causes other than suicide (MDD-NS), and the second one included elderly depressed patients who did not commit suicide (DEP). In the ACC region, MMD-S patients presented increased levels of *CRH* gene expression compared with controls and MDD-NS patients. No differences were observed in the DLPFCx of all groups [105].

The protein and gene levels of some targets related to the HPA-axis activity were also analyzed in another study by Perez-Ortiz and colleagues. Amygdala (AMY) samples were used to analyze changes in the *FKBP5* and *Nr3c1* in suicide victims compared with controls. Interestingly, the authors found a statistically significant reduction in the targets’ gene and protein expression in this brain region intensely involved in emotional regulation [106]. The gene expression changes in both targets were also analyzed in the PFC of 121 brain samples of victims who died by suicide and 88 non-suicidal victims. In this study, the authors only found a significant reduction in the *Nr3c1* transcript, and no differences were found in the case of the *FKBP5* [107]. Perhaps because FKBP5 acts as a cochaperone that modulates glucocorticoid receptor activity in response to stressors, a reduction in the gene encoding the receptor means that FKBPA is unchanged.

Despite the interest in exploring central brain markers associated with the stress axis and suicidal behavior after an early life adversity, from a clinical point of view, the peripheral markers are more appropriate because of the facility to access their measurement in patients. Therefore, searching for peripheral changes in different targets should be the first step of the approach to identify suicide-related biomarkers in patients with a previous history of early life stress and set up effective preventive strategies. In this line, the available information is scarce, and more studies are needed to deeply explore this association. 

## 5. Inflammation and Suicide

The relationship between inflammation and suicide has been widely explored, with heterogeneous results. The correlation between inflammation, suicide, and early life stress was evaluated in a study by Su and colleagues. Serum concentrations of inflammatory markers, HPA-axis markers, and metabolism were assessed for their potential association with suicidal behavior. Authors found higher concentrations of chemokine (C-X-C motif) ligand 1 (CXCL-1) in the serum of patients with suicide risk than non-suicidal patients. In these patients, the higher suicidal risk was associated with adulthood adversity and not early life stress [108]. 

Adverse childhood adversities are also strongly associated with self-harm behaviors. Russell and colleagues found a positive link between early life adversity and self-harm; however, no associations were reported about inflammatory biomarkers (interleukin (IL) 6 and C-reactive protein) [109]. The oxidative status has also been associated with suicidal behavior. Blood samples of patients with affective disorders and controls were analyzed, and alterations in the CMPAase/high-density lipoprotein (HDL) cholesterol levels were found in those exposed to early life stress. However, the direct correlation of these markers with suicide behaviors should be further investigated [110]. 

To assess the involvement of the immune system in suicide behavior in the early life stage, the gene and protein levels of the tumor necrosis factor-alpha (TNFα), IL-1β, and IL-6 were analyzed in the prefrontal cortex (PFC) of 24 teenage suicide completers and 24 matched controls. The results obtained in this study revealed a significant increase in the gene and protein expression of all three cytokines in the suicide victims, suggesting that these proinflammatory cytokines may have an essential role in suicide behavior [111]. In another study, the levels of different chemokynes and IL10 and IL16 in 16 suicide completers and 23 controls were evaluated in the PFC. The authors found a significant reduction in chemokines and IL10 levels (anti-inflammatory interleukine), whereas IL16 (pro-inflammatory interleukine) levels were increased in this brain region of suicide completers, supporting the critical role of the immune system in suicidal behavior [112].

Relative gene expression changes in different inflammatory markers involved in allergic processes (TNFα, IL-1β, IL-4, IL-5, IL-6, and IL-13) were analyzed in the orbitofrontal cortex of suicide completers with a history of major depressive disorder (MDD) versus controls. The authors reported increases in IL4 and IL13 gene expression in suicide completers compared with controls [113].

In blood samples, the plasma concentrations of different cytokines (IL-1α, IL-1β, IL-2, IL-4, IL-6, IL-8, IL-10, TNFα, epidermal growth factor (EGF), and vascular endothelial growth factor (VEGF)) were analyzed using an ELISA assay, in suicide attempters with a history of MDD. In this study, a significant association between lower VEGF levels in the plasma and suicide completion was found, suggesting the involvement of angiogenesis and immune response in suicide behavior [114]. In another study, IL2, IL6, and TNFα concentrations were measured in the plasma of suicide attempters, non-suicidal depressed patients, and healthy controls. Increased concentrations of IL6 and TNF-α and decreased levels of IL2 were found in suicide attempters compared with non-suicidal depressed patients and healthy controls, demonstrating that these markers of the immune system’s activity could act as distinguishing markers between suicide and non-suicide behavior in depressed patients [115]. 

The concentrations of VEGF and IL8 were also measured in the cerebral spinal fluid (CSF) of suicidal attempters versus controls. The authors found lower concentrations of VEGF, suggesting a lack of trophic neuronal support and reduced neurogenesis in the HIPP of the patients, indicating more severe depressive status. Interestingly, IL6 levels did not differ between all the evaluated groups [116]. However, IL6 concentrations were significantly increased in the CSF of suicide attempters compared with controls. They were even higher in patients with a violent phenotype of suicide attempt compared with non-violent phenotypes [117]. 

A critical molecular link between inflammation and suicide is the tryptophan metabolism via the kynurenine pathway. This pathway is responsible for 90% of tryptophan degradation. Several proinflammatory cytokines (IL1 beta and IL6, among others) activate this pathway, increasing the synthesis of quinolinic (QA) or kynurenic (KA) acid. This QA can induce excitotoxicity through NMDA glutamatergic receptor activation. This pathway has been analyzed in suicide completers and suicide attempters. In the postmortem anterior cingulate cortex of suicide victims with MDD and controls, the authors found a significant reduction in kynurenic acid levels in MDD suicide victims compared with controls and MDD patients who did not die by suicide [118].

In summary, despite the extensive available information about the inflammation involvement in suicide, the specific association of the changes in the levels of different interleukins in suicidal behavior is not clear because of the heterogeneous results. This heterogeneity could be associated with the type of sample used for the analysis (postmortem brain samples, plasma, and CSF) and psychiatric comorbidities in patients. Therefore, the challenge of finding patients with suicidal behavior without any additional psychiatric comorbidity hampers the search for specific suicide-related biomarkers.

## 6. Neuropeptides and Suicide

Peptide hormones, such as vasopressin and oxytocin, play a crucial role in the normal and pathological functions of the central nervous system. They are involved in stress response and adaptation; the first studies analyzing the involvement of these neuropeptides in emotional regulation date to the late 90s. In these studies, authors found increased vasopressin concentrations in the plasma and brain postmortem samples of depressed patients. An increased number of vasopressin and oxytocin-positive neurons was also reported in the paraventricular nucleus of the hypothalamus [119,120]. The plasmatic concentrations of vasopressin have been associated with hypercortisolemia and suicide attempts in depressed patients, supporting the available information about the involvement of this neuropeptide in the HPA-axis regulation [121,122]. In a study by Brunner and colleagues, the authors evaluated the plasma and CSF concentrations of neurotransmitters and peptides, including vasopressin. The authors did not find any alteration in these hormones between MDD patients (nine of them being suicide attempters) compared with controls. Only a positive correlation was found between vasopressin and cortisol levels [123].

Similarly, plasma vasopressin levels were assessed in suicide attempters and non-attempters after DMS. No differences were found in the vasopressin concentrations between both groups of patients after the test [124]. However, considering vasopressin-induced regulation of the HPA axis, more studies are required to explore its potential involvement in suicidal behavior further. 

Oxytocin is a nonapeptide that is intensely involved in childbirth and lactation, as well as social behavior, memory, and cognition. The peripheral concentrations of this neuropeptide were significantly reduced in suicide attempters compared with healthy subjects [125]. In addition, stress exposure, such as social exclusion, also decreased oxytocin concentrations in young adults with a suicide attempt history [126]. Oxytocin levels were also analyzed in the CSF and plasmatic samples of 28 drug-free suicide attempters and 19 healthy controls. The authors found lower CSF oxytocin levels compared to healthy volunteers, suggesting that this hormone may act as a predictor of suicide attempts [127]. These results indicate that oxytocin could by potentially useful as a biomarker of suicide behavior. However, more studies are needed to confirm this hypothesis. 

In a study conducted in the early nineties, brain postmortem samples of suicide victims and age-matched controls were analyzed to evaluate potential changes in the neuropeptide Y concentrations. The authors measured this neuropeptide in the frontal cortex, temporal cortex, caudate nucleus, and cerebellum. The results revealed significantly reduced neuropeptide concentrations in the frontal cortex and caudate nucleus. Interestingly, these reductions were more robust in suicide victims with a history of MDD [128]. In addition, reduced gene expression of neuropeptide Y was found in the prefrontal cortex of suicidal patients with a history of bipolar disorder [129]. Some additional studies were carried out to reveal the involvement of neuropeptide Y in suicide. The expression of this peptide, along with its four receptors, was analyzed at the transcriptional and translational levels in the PFC and HIPP of the postmortem brain of normal healthy and depressed suicide subjects. The authors reported a statistically significant decrease in the *neuropeptide Y* (*NPY*) gene expression and an increase in the gene expression of its receptors *NPY1R* and *NPY2R* in both brain regions. In addition, a decrease in the protein levels of the peptide in the PFC was also found in depressed suicide victims [130]. Neuropeptide Y has been proposed as a resilience-related peptide, and its involvement in different neuropsychiatric disorders was also evaluated. More studies are necessary to explore its precise role in suicide behavior and its potential usefulness as a biomarker of suicidality.

## 7. Serotonergic System Alteration in Suicide Victims

Psychiatric disorders associated with serotonergic dysfunction include schizophrenia and MDD. Several medications to treat these disorders modulate the serotonergic system [131]. Both diseases have a high incidence of suicide victims, and depression and suicide have been associated with reduced serotonergic transmission. In the dorsal raphe nucleus (DRN), where the primary neurons innervate the prefrontal cortex [132], a decrease in the total number of 5-HT1A receptors was observed in suicide cases compared to controls [133]. In addition, the suicide group had fewer DRN neurons expressing the *5-HT1A* gene than the control group [133]. An earlier study by Arango et al. (1995) found an increase in 5-HT1A receptor binding and a decrease in serotonin transporter binding in the ventrolateral prefrontal cortex in suicide cases, suggesting that alterations in the serotonin system occur primarily in these areas [134].

The protein TPH, involved in serotonergic neurotransmission, has been studied extensively. The gene for tryptophan hydroxylase is located at the 11P15.3-P14 locus. In 1994, Nielsen’s work was the first to report an association between suicide attempts and a polymorphism in intron 7 of the *TPH* gene [135]. Several years later, Turecki et al. (2001) found that one haplotype was more common in suicide completers than in controls, and this haplotype was also more common in patients who committed suicide by violent means [136]. However, further analysis did not reveal differences between controls and victims for the *TPH1* gene at the single locus level nor the haplotype level in men of Slavic origin [137]. In a Japanese population, no significant association was found between *TPH* and *5-HT1A* in suicide samples compared to controls [138]. In another study, *TPH2* gene expression was significantly increased in the ventral prefrontal cortex of suicide completers compared to controls, but this could not be associated with the polymorphism SNPrs10748185 located in the promoter region of *TPH2* and suicide [139].

The serotonin transporter (5-HTT) is responsible for the sodium-dependent serotonin reuptake, and the *5-HTT* gene is located at the 17q11.1-q12 locus. Postmortem studies have found a significant frequency of the long allele of the *5-HTT* gene in depressed suicide victims [140]. In 2003, an increase in the 5-HTTLPR L allele was found in the suicide group [141]. However, in the Chinese population, no association was found between changes in the *5-HHT* genes and suicidal behavior [142]. Given these inconclusive findings, attributing suicidal behavior solely to genetic alterations in this system is overly simplistic. However, the serotonergic system is involved in both the predisposition to suicidal behavior related to decision-making and as a significant stressor [143]. Suicidal behavior is also intricately linked to genetic, environmental, and gender factors.

## 8. Endocannabinoid System Alteration in Suicide Victims

The endocannabinoid system (ECS) has emerged as a potential target in psychiatric disorders because it regulates emotional responses. The ECS also regulates cognitive processes, inflammation, chronic pain, and epilepsy [144,145]. Studies in rodents have shown that cannabinoid receptors are critical in responding to stress, anxiety, and depression [146,147,148,149,150,151,152,153]. A higher frequency of single nucleotide polymorphisms (SNP) of the CB1r gene (*CNR1*) was found in patients with MDD [154,155] and schizophrenia [156]. 

Based on previous animal studies, the CB2r in the central nervous system has a role as a possible anxiolytic and antidepressive target [157,158,159]. G protein-coupled receptor 55 (GPR55) is another target of endocannabinoids [160], with preliminary results suggesting that early life stress induces changes in GPR55 gene expression in mice [161]. Postmortem studies have shown that CB2r and GPR55 gene expression was significantly lower in the DLPCx of suicide victims, although CB2r protein expression was higher in said suicide victims. The presence of CB2-GPR55 receptor heteromers was also found in both neurons and astrocytes [162].

Curiously, an alteration in CB1r was also found in the DLPCx of suicide victims, with a higher CB1r density in alcoholics who died by suicide than in the control group, as confirmed by Western blot analysis. In addition, higher levels of anandamide (AEA) and 2-arachidonoylglycerol (2-AG) were found in the DLPCx of AS [163] (Figure 1). The same group has investigated changes in CB1r in the ventral striatum in alcohol-dependent suicides and found an upregulation of CB1r and an alteration of the FAAH enzyme in the ventral striatum [164]. Furthermore, CB1r gene expression was increased in the DLPCx of depressed suicide victims compared to controls [165]. A cross-sectional study conducted in 2020 found elevated levels of AEA and N-palmitoylethanolamine (PEA) in the serum of suicide attempters compared to controls [166]. Therefore, some evidence suggests that endocannabinoid system dysfunction (ECS) plays a relevant role in the pathophysiological aspects of suicidal behavior [167,168]. 

## 9. Excitatory and Inhibitory Balance Alteration in Suicide

Gamma-aminobutyric acid (GABA) and glutamate (Glu) are the mammalian central nervous system’s primary inhibitory and excitatory neurotransmitters and have been found to be involved in suicidal behavior. Alterations in excitatory and inhibitory neurotransmission have been extensively studied in patients with MDD who have or have not committed suicide. Notably, in 2004, Merali et al. showed that the gene expression of *CRH* and *GABAA* in the PFC were altered in depressed suicides, consistent with changes in the regulation of the HPA-axis in suicide cases [104]. Other studies have also demonstrated disturbances in the balance between excitatory and inhibitory processes in suicide. In 2005, Jones’ group identified the significant downregulation of *SLC1A2* and *SLC1A3*, critical members of the glutamate-neutral amino acid transporter protein family, as well as L-glutamate ammonia ligase, an enzyme responsible for converting glutamate to nontoxic glutamine [169]. These alterations could significantly increase extracellular glutamate levels, potentially leading to neurotoxicity and reduced signaling efficiency. In 2023, our research group showed a decrease in vesicular GABA transporter (VGAT) immunoreactivity and an increase in vesicular glutamatergic transporter-1 (VGluT1) immunoreactivity in the HIPP of individuals who died by suicide [170].

Using microarray analysis and RT-PCR, Turecki’s group showed in 2009 that glutamatergic and GABAergic-related genes were globally altered in prefrontal cortical areas and the HIPP in suicide cases [171]. A protein-coding transcript (ENST00000414552) of the GABA-A receptor, gamma 2 (*GABRG2*), presented lower brain expression in postmortem suicides and was significantly decreased in the ACC in mood disorders [172,173]. Consistent with these results, a recent RT-PCR analysis of 32 genes encoding proteins directly involving glutamatergic or GABAergic synaptic transmission revealed the significantly increased expression of genes associated with these processes in the ACC in individuals with MDD who died by suicide (MDD-S). Conversely, in the DLPFCx, the expression of these genes was decreased in MDD-S cases [174].

In conclusion, several factors and systems are altered in individuals who die by suicide (Table 1). Further research is needed to identify biomarkers that can serve as early warning signs for healthcare professionals and help prevent suicidal behavior.

## 10. Epigenetic Changes in Suicidal Behavior

Epigenetic studies examine the interaction between the environmental and genetic risk factors of suicide. Epigenetic alterations include DNA methylation, histone acetylation and non-coding RNAs, which control gene expression without affecting the DNA sequence. DNA methylation is one of the most studied epigenetic modifications. It is regulated by DNA methyltransferases (DNMTs), involving the addition of a methyl (-CH3) group to the fifth carbon of a cytosine–guanine (CpG) dinucleotide. This epigenetic modification is associated with the repression of gene expression [175].

The evidence achieved to date identified DNA methylation alterations in genes involved in stress response and neuronal plasticity in the peripheral blood and brain regions of psychiatric patients who died by suicide, with previous suicide attempts or suicidal ideation (Figure 3).

### 10.1. DNA Methylation Alterations in Peripheral Blood Tissues

DNA methylation changes have been found in the peripheral blood tissues of patients with suicidal ideations and behaviors. One of the most well-studied genes in the epigenetics of suicide is the *NR3C1* gene, which encodes the glucocorticoid receptor (GCr) [176]. This receptor is involved in cortisol signaling, stress, and inflammatory responses [177,178]. Abnormalities in the methylation status of *NR3C1* impact the regulation of the negative feedback loop of the HPA-axis, inducing a poorer adaptation to stress [179]. In lymphocytes from women with bulimia nervosa and comorbid bipolar disorder (BPD) or a history of suicidality, higher methylation was found in specific GR exon 1C promoter sites [180]. In addition, the hypermethylation of promoters of *NR3C1, FK506* binding protein 5 (FKBP5), and CRHBP genes, additional critical elements of the HPA axis, were found in major depressive MDD patients with and without suicidal ideation. Interestingly, these genes’ methylation degree was positively correlated with severe suicidal ideation [181]. Moreover, epigenetic changes in the *CRH* gene (hypomethylation of two CpG CRH cg19035496 and cg234090749) were found in adult suicide attempters. Curiously, the hypermethylation of cg19035496 was linked with a high general psychiatric risk score in adolescents [182]. 

*SKA2* is another gene that regulates GCr nuclear transactivation [183,184]. Epigenetic SKA2 gene variations have been proposed to impact the modulation of HPA-axis function [183], and the increased methylation of SKA2 was identified in the blood of suicide victims [185]. 

DNA methylation alterations, which are involved in neuroplasticity, have also been found in the BDNF. Higher methylation levels of the *BDNF* promoter were identified in MDD patients with suicidal ideation compared to those without [186]. Similarly, higher *BDNF* promoter methylation levels were significantly associated with a previous history of suicide attempts, suicide ideation during treatment, higher Beck Scale for Suicide Ideation (BSS) scores, and poor treatment outcomes for suicidal ideation [187]. In addition, the hypermethylation of *BDNF* promoters was found in MDD patients with and without suicidal ideation. The degree of methylation was positively correlated with severe suicidal ideation [181]. In agreement with these findings, a recent study contributes to establishing a connection between the increased methylation of *BDNF* and severe suicidal behavior in women [188].

Similarly, increased *BDNF* methylation was associated with suicidal ideation and depression one year after breast surgery in women with breast cancer [189]. However, hypomethylation of the *BDNF* region upstream of exon I has been found in blood samples of suicide victims. In contrast, no differences in DNA methylation were observed in the hippocampus or the DLPFCx [190]. 

Some studies revealed epigenetic alterations in genes coding for the critical elements of the monoaminergic system. In the case of serotonin receptor 1A (*HTR1A*), hypomethylation was found in the blood of suicide victims. For amplicon monoamine oxidase A (MAOA) gene exon I (*MAOA_1*), the results showed increased blood methylation in suicide victims [185]. In female affective disorder patients with a history of violent suicide attempters, the decreased methylation of the MAOA gene exon I promoter region was identified in DNA samples of peripheral blood cells [191]. The blood methylation status of the *SCL6A4* gene, which encodes the serotonin transporter, has been associated with suicidal ideation in stroke patients [192]. Zhang and colleagues evaluated the Tryptophan Hydroxylase *TPH2* methylation, an enzyme involved in serotonin synthesis [193], in peripheral blood samples in patients diagnosed with MDD who had attempted suicide and control patients with MDD. The results revealed that *TPH2* was hypermethylated in MDD with suicide. This epigenetic change was also associated with reduced *TPH2* mRNA levels [194]. However, in the study by Kouter and colleagues, hypomethylation of TPH2 was found in the blood of suicide victims [185]. 

Recently, differences in the methylation of the Cyclin-Dependent Kinase 5 (CDK5) gene have been found in the PMBCs of US military veterans with suicide attempts [195]. *CDK5* is highly expressed in the adult brain and is relevant in regulating neuronal growth, survival, learning, and memory [196]. Interestingly, significant differences in the expression of *CDK5* were identified in the PFC of individuals who died by suicide [197].

GWASs provide relevant information regarding a different blood methylation profile in suicide. Such is the case of the study by Jeremian and colleagues, who found a different blood methylation profile between BPD patients with and without suicidal behaviors. More specifically, there is less methylation in the 5′ untranslated region of Membrane palmitoylated protein 4 (MMP4) and intron 3 of TRE2/BUB2/CDC16 domain family member 16 (*TBC1D16*) and less methylation in exon 1 of nucleoporin 133 (*NUP133*) in BPD patients with a history of suicidal behaviors [198]. 

In addition, hypomethylation in the CpG site cg19647197 within the *CCDC53* gene was found in the white blood cells of schizophrenic suicide attempters [199], which is a core subunit of the WASH complex involved in endosomal fission [200]. 

Finally, DNA methylation in regulatory regions of the Elovl5 gene (elongation of very long-chain fatty acids protein 5) was associated with the diagnosis of MDD and suicide attempts. This epigenetic change explains the connection between plasma polyunsaturated fatty acid (PUFA) levels and suicide risk [201].

### 10.2. DNA Methylation Alterations in the Brain from Postmortem Studies 

Postmortem studies have identified DNA methylation alterations in different brain regions of individuals who died by suicide. As described in blood peripheral cells, hypermethylation of the *NR3C1* gene was detected in the HIP of individuals who died by suicide with a history of childhood abuse [202]. Moreover, decreased levels of mRNA, mRNA transcripts bearing the glucocorticoid receptor 1F splice variant, and increased cytosine methylation of the *NR3C1* promoter have been identified in the same samples [203]. Recently, lower methylation levels of the 1B promoter have been determined in the HIP of suicide victims, where methylation levels were higher in the insula and blood. On the other hand, mixed methylation patterns were found in the amygdala and Broadman area 46 [185]. Complementarily, Rizavi and colleagues found a strong correlation between the DNA methylation status and gene expression changes in specific *NR3C1* exon-1 variants in the PFC of teenage suicide completers. Moreover, in the HIP and PFC, the aberrant expression of DNA methylating and demethylating enzymes was also identified [204]. 

An additional gene evaluated was SKA2 due to its connection with GCr. The opposite methylation status was found in suicide victims, depending on the brain region analyzed. Decreased *SKA2* methylation was found In the HIP, whereas increased methylation was identified in the insula [185]. A complementary study on individuals with high interpersonal violence and psychosocial stress rates showed that *SKA2* DNA methylation variations mediate vulnerability to suicidal behaviors and PTSD through the dysregulation of the HPA axis in response to stress [205]. Furthermore, an altered methylation status at *SKA2* and three additional CpGs (ATP8A1, LOC153328, and KCNAB2) were found in the frontal cortex of suicide victims [206]. Curiously, a strong correlation has been observed with the *NR3C1* and *SKAAK2*, since suicide victims often have increased DNA methylation of one or the other [207].

Analysis of the methylation status of the BDNF promoter received particular interest due to its role in neuroplasticity. Higher levels of methylation in the BDNF promoter were identified in the Wernicke area of suicide victims compared to controls. Interestingly, this methylation status was associated with lower mRNA expression [208]. The same authors carried out one study analyzing the methylation status and gene expression of the tropomyosin-related kinase B (*NTRK2*), one of the receptors of BDNF, without detecting differences [209]. Curiously, the methylation state of the promoter region of the truncated variant of the TrkB receptor (*TrkB.T1*) was associated with its expression in the frontal areas of individuals who died by suicide [210]. In addition, hypermethylation of the TrKB-T1 3’UTR region was found in the frontal cortex (BA8/9) of suicide completers, showing an association with reduced *TrKB-T1* expression [211]. These epigenetic alterations could be underlying the plasticity changes observed in suicide completers.

Recently, another study evaluated potential epigenetic alterations in genes coding critical elements of the serotoninergic system, the *SLC6A4* and the *HTR1A*. Hypomethylation of the *SLC6A4* gene has been identified in the BA46 of suicide victims. On the contrary, the methylation pattern was mixed in the other measured tissues (HIP, amygdala, and insula). In the case of *HTR1A*, the study revealed hypermethylation in the insula and mixed methylation in the HIP, AMY, and BA46 [185]. In the same study, epigenetic alterations were analyzed in the *TPH2*, observing hypomethylation in the BA46 and hypermethylation in the HIP. Moreover, differences in the methylation of MAOA, an enzyme involved in serotonin degradation [212], were found in the HIP, insula, AMY, and Brodmann area 46 of suicide victims [185]. Altogether, these results revealed epigenetic alterations in critical elements of the monoaminergic system. 

Furthermore, DNA methylation alterations have also been analyzed in GABA receptors. Alterations in the functioning of the inhibitory GABAergic signaling have been proposed as one of the etiopathogenic mechanisms underlying different neuropsychiatric disorders, including suicide. Hypermethylation of the GABAA receptor alpha 1 subunit promoter region was found in the frontopolar cortex of individuals who committed suicide and were diagnosed with MDD [213]. In line with this finding, a recent study found differences in the methylation status for the amplicons *GABRA1_1* and *GABRA1_2* [185]. 

Finally, GWAS methylation studies provide relevant information about altered methylation patterns in different brain regions of suicidal samples. In the HIP, Labonte and colleagues found differences in the methylation patterns of male suicide victims, the majority of them hypermethylated and closely related to cognitive processes, such as NR2E1, which encodes a brain-specific orphan nuclear receptor acting as a transcriptional repressor [214]; *GRM7*, which codes for the G-protein coupled metabotropic glutamate receptor subunit 7 [215]; *CHRNB2*, encoding a brain-specific subunit (β2) of the ligand-gated ionotropic nicotinic acetylcholine receptor family [216,217]; and *DBH*, coding for a catecholamine-synthetic membrane-bound enzyme responsible for the synthesis of norepinephrine [218]. Interestingly, promoter methylation differences were inversely correlated with gene expression differences [202].

In the PFC of male death by suicide and controls, Schneider and colleagues identified differences in methylation patterns between groups studying single CpGs over the genome. The study highlights the enrichment of the genes *APLP2* (encodes the amyloid precursor-like protein 2), *BDNF*, *HTR1A* (encodes the serotonin receptor 1A), *NUAK1* (encodes the enzyme AMPK-related protein kinase 5), *PHACTR3* (encodes a member of the phosphatase and actin regulator protein family), *MSMP* (encodes the prostate-associated microsemiprotein), *SLC6A4*, *SYN2* (synapsin 2), and *SYNE2* (codes the Spectrin Repeat Containing Nuclear Envelope Protein 2) and supports a role for epigenetics in the pathophysiology of suicide [219]. Gaine and cols. found that 55% of the differentially methylated regions (DMRs) were hypermethylated in the PFC of suicidal samples. The genes found to be altered and methylated belonged to several pathways, including axonal guidance signaling, calcium signaling, β-adrenergic signaling, and opioid signaling [220]. Similarly, altered DNA methylation patterns (affecting 4430 genomic regions) were found in the PFC of males who died by suicide, identifying 10 genes with potential relevance to suicide and psychiatric disorders involved in synaptic plasticity, HPA-axis, and additional pathways [221]. Complementary studies found different methylation levels in the PFC of suicide victims [222,223]. Interestingly, Cabrera-Mendoza and colleagues found a significant correlation between 22CpGs and gene expression in suicide victims [224].

Kouter and colleagues examined alterations in the pattern of DNA methylation in the HIP and PFC of suicide victims, finding differences in the methylation levels of approximately 3000 CpGs. The gene analyses provide information about enrichment for genes associated with cell structural integrity and nervous system regulation [225].

In an additional GWAS study, hypomethylation in the intronic region of the *ELAVL4* gene was found in the PFC of individuals who died by suicide. This gene plays a role in the translation and stabilization of mRNA, modulating neuronal development and maintenance. In the cerebellum, the altered methylation status of six genes related to long-term synaptic depression has been found between suicide completers and controls [11]. 

In summary, several studies on DNA methylation have been conducted in recent years, contributing to a better understanding of the epigenetic mechanisms underlying suicide. These studies revealed DNA methylation alterations in the serotoninergic system, HPA and neuroplasticity markers. However, it is hard to achieve conclusive results, mainly due to the differences in the methodology employed by each study and the sample analyzed. 

## 11. Conclusions

This article reviews the molecular changes occurring in individuals with psychiatric disorders or those undergoing epigenetic modifications due to adversity, precipitating suicidal behavior. 

Individuals struggling with a psychiatric disorder such as MDD, BP, or schizophrenia are more likely to manifest suicidal behavior. Alterations in the serotonergic system, HPA-axis, GABAergic/glutamatergic system and BDNF reduction were reported in BP, MDD and schizophrenic patients. Increased proinflammatory cytokines and changes in the HPA-axis were also detected in patients with AD. In the case of BPD, the inclination towards suicidal behavior is often intertwined with comorbidities. Nevertheless, impulsive and unforeseen suicide cases may arise from an inadequate ability to cope with stressful situations. Indeed, early-life adversity may induce heightened stress and anxiety, ultimately leading to the modification of the HPA-axis, potentially escalating susceptibility to suicidal tendencies (Figure 2).

Suicide victims exhibit diverse central and peripheral alterations, irrespective of a prior psychiatric diagnosis. Notably, DNA methylation changes have been documented in the peripheral blood tissues of individuals with suicidal ideation and behavior. For instance, epigenetic alteration in the methylation status of the *NR3C1* gene, which encodes the GCr [176], which is involved in cortisol signaling, stress, and inflammatory responses [177,178], was found in individuals with childhood abuse or exposure to a highly stressful situation that may trigger PTSD later [205]. Also, genetic alterations in this serotonergic system have been found in suicide victims. 

Other systems are also affected by suicidal behavior. The study of the ECS has been of great importance in recent years. Postmortem studies have shown that CB2r and GPR55 gene expression was significantly lower in the DLPCx of suicide victims, although CB2r protein expression was higher in suicide victims. In addition, several alterations in the GABAergic and glutamatergic balance, as well as an increase in the inflammatory interleukins IL-1, IL-6, and TNF-alpha and a decrease in BDNF, were found in the PFC of suicide victims. 

In summary, suicidal behavior may result from a confluence of factors. It is imperative to develop a comprehensive, multifaceted suicide prevention strategy highlighting the severity of suicide as a global and preventable issue.

## 12. Methods

The literature review consisted of an exhaustive search for scientific information in the Medline database (PubMed). To identify molecular alterations and risk factors of suicide, the following keywords and their combinations were used: “major depression disorder” AND “molecular alteration” AND “suicide”, “bipolar disorder” AND “molecular alteration” AND “suicide”, “borderline personality disorder” AND “molecular alteration” AND “suicide”,” anxiety disorder” AND “molecular alteration” AND “suicide”, “schizophrenia” AND “molecular alteration” AND “suicide”, “substance use disorder” AND “molecular alteration” AND “suicide”, “early life adversity” AND “suicide”, “inflammation factor alteration” AND suicide”, “neuroendocrine system alteration” AND “suicide”, “serotonergic system” AND “suicide”, “endocannabinoid system alteration” AND “suicide”, “GABAergic system alteration” AND “suicide” “Glutamatergic system alteration” AND “suicide” “Epigenetic changes” AND “suicide”. To search gene alteration the following keywords and their combinations were used: “major depression disorder” AND “gene alteration” AND “suicide”, “bipolar disorder” AND “gene alteration” AND “suicide”, “borderline personality disorder” AND “gene alteration” AND “suicide”,” anxiety disorder” AND “gene alteration” AND “suicide”, “schizophrenia” AND “gene alteration” AND “suicide”, “substance use disorder” AND “gene alteration” AND “suicide”. AND “screening tools” were combined with terms related to the technical approach using the Boolean operator “AND”; “molecular changes”, “gene alteration” AND “suicide”. References of identified publications were included in the additional searches. Articles published in predatory journals were excluded from the screening process.

## Figures and Tables

**Figure 1 ijms-24-16726-f001:**
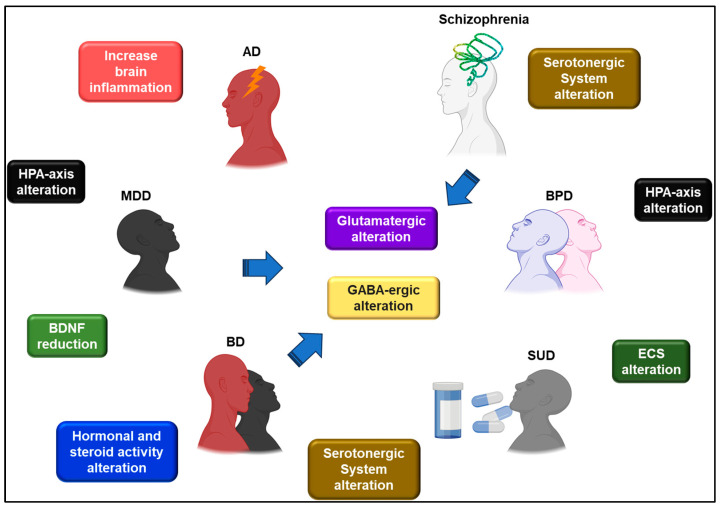
**Molecular changes found in the brains of suicide victims with psychiatric disorders.** In BP, alterations in the serotonergic system, the HPA-axis in the GABAergic system, and BDNF reduction have been found. In MDD, alterations in the HPA-axis, GABAergic-Glutamatergic system, BDNF reduction, and increased proinflammatory cytokines were detected. Increased proinflammatory cytokines and changes in the HPA-axis were found in AD. In schizophrenia, alterations in the serotonergic system, the HPA- axis, and the glutamatergic system were reported. In BPD, the changes found are associated with comorbidity with other disorders. Finally, alterations in the ECS have been found in the brains of alcoholics who committed suicide. AD: anxiety disorder, BD: bipolar disorder, BDNF: brain-derived neurotrophic factor, BPD: borderline personality disorder, ECS: endocannabinoid system, MDD: major depression disorders, SUD: substance use disorder.

**Figure 2 ijms-24-16726-f002:**
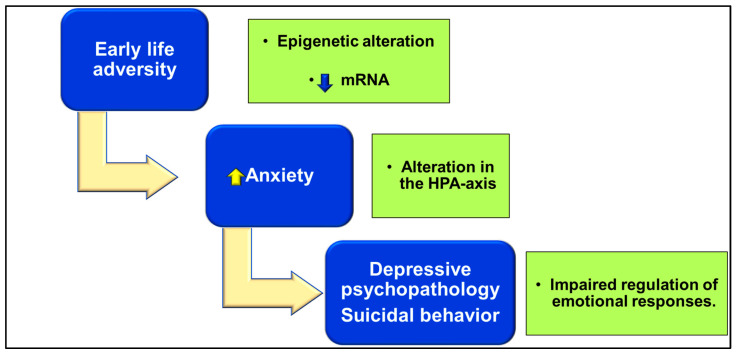
**Flowchart linking early life adversity and suicidal behavior.** up arrow: increase; down arrow: decrease.

**Figure 3 ijms-24-16726-f003:**
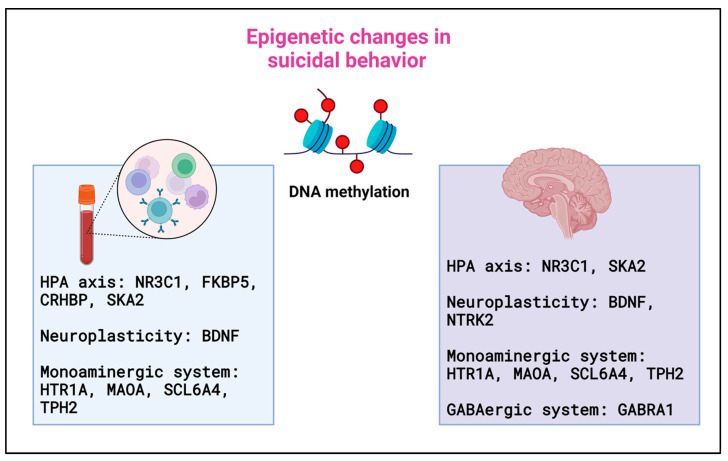
**Main epigenetic changes in suicide behavior.** The figure represents the main DNA methylation changes observed in peripheral blood cells and brain areas of patients with suicide behavior and in those who died by suicide, respectively. HPA axis: hypothalamus–pituitary–adrenal axis, NR3C1: gene encoding glucocorticoid receptor, FKBP5: the gene encoding the FK506-binding protein 5, CRHBP: the gene encoding the corticotropin-releasing hormone binding protein, SKA2: gene encoding spindle and kinetochore associated complex subunit 2, BDNF: the gene encoding the brain-derived neurotrophic factor, HTR1A: the gene encoding serotonin receptor 1 A, MAOA: the gene encoding monoamine oxidase A, SCL6A4: serotonin transporter gene, TPH2: gene encoding tryptophan hydroxylase 2, NRTK2: the gene encoding neurotrophic receptor tyrosine kinase 2; GABRA1: the gene encoding the alpha-1 subunit of the GABAA receptor protein.

**Table 1 ijms-24-16726-t001:** Main system alteration found in human samples of suicide attempters and completers.

System Affected	Change	Sample (Males:Females) (M:F)	References
**HPA axis**	↓ Plasma cortisol post-DST	MDD (28F) and MDD plus self-harm (29F)	[99]
	↓ Plasma cortisol post-stress levels	Suicide attempt (47, 17M:30F), suicidal ideation without attempt (53, 24M:29F)	[100]
	No correlation with 24-h urinary cortisol levels	General population 588F, 506M	[101]
	↓ Baseline plasma cortisol levels and ↑ after stress exposure	MDD (126F)	[102]
	↓ BDNF in ACC of suicide victims ↓ BDNF in ACC of ELA patients	ELA (13M), RLS (13M)	[103]
	↑ CRH in FPC and DMPFC ↓ CRH1 gene expression in FPC	Suicide completers with MDD (11M:1F)	[104]
	↑ CRH and NIDD in ACC	Suicide with MDD (17, 10M:7F)	[105]
	↓ FKBP5 and GR gene and protein expression in AMY	Suicide completers (13M)	[106]
	↓ GR transcript expression	Suicide completers (121M:F)	[107]
**Inflammatory markers**	↑ CXCL-1 in serum	Suicide risk and MDD (50, 15M:35F)	[108]
	No changes in inflammatory markers	The general population with ELA and self-harm (4308F)	[109]
	No changes in oxidative status	Affective disorders (118, M:F)	[110]
	↑ TNFα, IL-1β and IL6 gene and protein levels in PFC	Suicide completers (24, 14M:10F)	[111]
	↓ Chemokines and IL10 and ↑ IL16 in PFC	Suicide completers (16, 10M:6F)	[112]
	↑ IL4 and IL13 gene expression in OFC	Suicide completers (34, 20M:14F)	[113]
	↓ VEGF in plasma	Suicide attempts (58, 23M:25F)	[114]
	↑ IL6 and TNFα and ↓ IL2 in plasma	Suicide attempts (47, 21M:26F)	[115]
	↓ VEGF in CSF	Suicide attempts (43: 15M:28F)	[116]
	↑ IL6 in CSF	Suicide attempts (63, M:F)	[117]
	↓ KA in ACC	Suicide completers with MDD (44)	[118]
**Neuropeptides**	↑ Plasma vasopressin and hypercortisolism	Suicide attempts (45, 18M:27F)	[121]
	↑ Plasma vasopressin and cortisol	MDD (19, 7M:12F) being 9 suicide attempts	[123]
	No changes were observed in the vasopressin levels	Suicide attempts and MDD (13, 9M:4F), non-attempts and MDD (15, 10M:5F)	[124]
	↓ Serum oxytocin levels	Suicide attempts (48, 18M:30F)	[125]
	↓ Plasma oxytocin levels after stress exposure	Suicide attempts (100, 30M:70F)	[126]
	↓ CSF oxytocin levels	Suicide attempts (28, 18M:10F)	[127]
	↓ Neuropeptide Y in frontal cortex and caudate nucleus	Suicide completers (26, 21M:5F)	[128]
	↓ Neuropeptide Y gene expression in the PFC in BD	Suicide completers (SZP 15, 9M:6F; BD 15, 9M:6F; MDD 15, 9M:6F)	[129]
	↓ Neuropeptide Y and ↑ NPY1R and NPY2R gene expression in PFC and HIPP	Suicide completers with MDD (24, 14M:10F)	[130]
**Serotonergic system**	↓ 5-HT1A binding and total number in the DRN ↓ 5-HT1A gene expression in the DRN	Depressed suicide victims (6M:4F)	[133]
	↑ 5-HT1A binding in the PFC ↓ 5-HT1A binding in the PFC	Suicide completers (14M:8F)	[134]
	↑ TPH I7779C (L) allele frequency	Alcoholic violent offenders (185M)	[135]
	No association of the promoter (-7065CT) and intron 7 (218AC) polymorphisms of TPH1 gene	Violent suicide (160M)	[137]
	No association between A779C and A218C in the intron of TPH gene No association of Pro 16Leu in the coding region of 5-HT1A	Suicide victims in the Japanese population (95M:39F)	[138]
	No changes in SLC6A4 and TPH1 mRNA levels in VPFC ↑ TPH2 gene expression in VPFC No association of SNPs10748185 in the promoter region of TPH2	Suicide victims (27M:13F)	[139]
	↑ Frequency long allele of 5-HT1A	Depressed suicide victims (34M) Suicide victims (102M:33F)	[140] [141]
	No association of SNPs in 5-HT2A, 5-HT2C, 5-HTT	Suicide attempters (127M:126F)	[142]
**Endocannabinoid System**	↓ CB2r and GPR55 gene expressions in the DLPFCx ↑ CB2r protein in the DLPFCx ↑CB2r-GPR55 heteromers in the DLPFC	Suicide completers (18M)	[162]
	↑ CB1r protein in DLPFCx ↑ AEA and 2-AG in DLPFCx	Alcohol-dependents who died by suicide (7M:11F)	[163]
	↑ CB1r protein and binding, and FAAH activity in the ventral striatum	Alcohol-dependents who died by suicide (9M)	[164]
	↑ CB1r density and binding in DLPFCx	MDD who died by suicide. (24M)	[165]
	↑ AEA and PEA in serum	Suicide attempters (8M:22F)	[166]
**GABAergic** **System**	GABAA gene expression (alpha1, alpha3, alpha4, and delta receptor subunits) in PFC	Suicide completers (11M:1F)	[104]
	↓ VGAT-ir in HIPP	Suicide completers (28M)	[170]
	Alterations in GABAergic-related genes in PFC and HIP	Suicide completers with and without MDD (16:10)	[171]
	↓GABRG2 in DLPFCx	Suicide completers with MDD (13M:8F)	[173]
	↑ GABAergic-related genes in ACC ↓ GABAergic-related genes in DLPFCx	MDD suicide completers (10M:7F)	[174]
**Glutamatergic** **System**	mGluR5 gene expression in the AMY and HIPP ↑ VGluT1-ir in HIPP	Suicide completers (28M)	[170]
	Alterations in glutamatergic-related genes in PFC and HIP	Suicide completers with and without MDD (16:10)	[171]
	↑ Glutamatergic-related genes in ACC ↓ Glutamatergic-related genes in DLPFCx	MDD suicide completers (10M:7F)	[174]

↑: increase; ↓: decrease; 2-AG: 2-arachidonoylglycerol; ACC: anterior cingulate cortex; AEA: anandamide; AMY: amygdala; BD: bipolar disorder; CB1r: cannabinoid receptor 1; CB2r: cannabinoid receptor 2; CRH: corticotropin-releasing hormone;CRH1: corticotropin-releasing hormone receptor 1; CSF: cerebrospinal fluid; CXCL-1: chemokine ligand 1; DLPFCx: dorsolateral prefrontal cortex; DMPFC: dorsomedial prefrontal cortex; DRN: dorsal raphe nucleus; DST: dexamethasone suppression test; ELA: early life stress; FAAH: fatty acid amide hydrolase; FKBP5: FK506-binding protein 5; FPC: frontopolar cortex; GABAA: gamma aminobutyric type A receptor; GABRG2: gamma-aminobutyric acid type A receptor subunit gamma 2; GPR55: G protein-coupled receptor 55; GR: glucocorticoid receptor; HIPP: hippocampus; 5-HT1A: serotonin receptor 1A; 5-HT2A: serotonin receptor 2A; 5-HT2C: serotonin receptor 2C; 5-HTT: serotonin transporter; IL-1β: interleukine 1-beta; IL-2: interleukine 2; IL-6: interleukine 6; KA: kynurenic acid; MDD: major depressive disorders; mGluR5: metabotropic glutamate receptor subtype 5; NIDD: NOS1-interacting DHHC domain-containing protein with dendritic mRNA; NPY1R: neuropeptide Y receptor 1; NPY2R: neuropeptide Y receptor 2; OFC: orbitofrontal cortex; PEA: palmitoylethanolamide; PFC: prefrontal cortex; RLS: recent life stress; SNPs: single nucleotide polymorphisms; SLC6A4: gene that codifies serotonin transporter; SZP: schizophrenia; TNFα: tumor necrosis factor-alpha; TPH: tryptophan hydroxylase; TPH2: tryptophan hydroxylase 2; TPH1: tryptophan hydroxylase 1; VEGF: vascular endothelial growth factor; VGAT-ir: vesicular GABA transporter immunoreactivity; VGluT1-ir: vesicular glutamate transporter 1 immunoreactivity; VPFC: ventromedial prefrontal cortex.

## Data Availability

Not applicable.

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
