# Peer review of "Molecular Changes Associated with Suicide"

_ijms, 2023, doi:10.3390/ijms242316726_

Round 1

Reviewer 1 Report

Comments and Suggestions for Authors

Dear authors,

I want to congratulate for this subject. Suicide is a serious global public health problem. Many individuals do not disclose suicidal intentions despite frequent contact with health care professionals. Major depressive disorder (MDD) is the most common psychiatric diagnosis for suicide. Current studies demonstrate that suicide is associated with a dysregulated stress response system and an increase in inflammatory responses and their related effects. Stress response changes, including polyamine metabolism, circadian rhythm, immune dysregulation, and telomere maintenance are currently implicated in the etiopathogenesis of suicide. Future studies in large clinical populations investigating these systems may help identify suicidal MDD patients. So it is very important for the healthcare provider to know by specific methods, which patients can develop this risk.

Back to the article:

- the introduction is comprehensive, like the main text.

- the conclusions are exhaustive, they could be synthetically reorganized

- the method is presented too succinctly.

- the references are comprehensive

Overall, it is an excellent review in this topic.

Reviewer 2 Report

Comments and Suggestions for Authors

The authors propose a review on the molecular changes associated with suicide. The review is interesting and exhaustive and is certainly worthy of publication, after some revisions.

  1. The paper specifically deals with the genetic and molecular aspects of suicide: this focus must never be lost. If there is a risk of too extensive (sometimes distracting) discussion, it would be preferable to reduce the analysis of the other aspects (psychological, sociodemographic), although these issues also certainly add value to the work.
  2. In paragraph 2.3 the genetic aspects of the correlation between anxiety and suicidal behavior are not treated with the same depth as found in the other paragraphs. An in-depth study on this point would be appropriate. If the literature on this subject is poor, it would be necessary to point it out and, in any case, report the state of the art as far as possible.
  3. In paragraphs 2.5 and 2.6 there is no discussion concerning aspects of genetics and molecular biology. As in the previous point, if the literature in this regard is scant, it would be necessary to clarify it and, in any case, report the state of the art as far as possible.
  4. Particularly paragraphs 4 and 5 appear long and detailed, which can be certainly a positive thing for such a review, it could probably be an improvement to add a graph/table for each paragraph, in which the fundamental aspects are summarized. Paragraphs 6,7,8,9 (although certainly shorter) are also well suitable for a similar summary (by means of a graph or table). It would be appropriate to evaluate a schematization in this sense for them too.
